# Synthetic mammalian pattern formation driven by differential diffusivity of Nodal and Lefty

Ryoji Sekine[1], Tatsuo Shibata [2] & Miki Ebisuya [1,3]

A synthetic mammalian reaction-diffusion pattern has yet to be created, and Nodal-Lefty signaling has been proposed to meet conditions for pattern formation: Nodal is a short-range activator whereas Lefty is a long-range inhibitor. However, this pattern forming possibility has never been directly tested, and the underlying mechanisms of differential diffusivity of Nodal and Lefty remain unclear. Here, through a combination of synthetic and theoretical approaches, we show that a reconstituted Nodal-Lefty network in mammalian cells spontaneously gives rise to a pattern. Surprisingly, extracellular Nodal is confined underneath the cells, resulting in a narrow distribution compared with Lefty. The short-range distribution requires the finger 1 domain of Nodal, and transplantation of the finger 1 domain into Lefty shortens the distribution of Lefty, successfully preventing pattern formation. These results indicate that the differences in localization and domain structures between Nodal and Lefty, combined with the activator-inhibitor topology, are sufficient for reaction-diffusion patterning.

[1] Laboratory for Reconstitutive Developmental Biology, RIKEN Center for Biosystems Dynamics Research (RIKEN BDR), 2-2-3 Minatojima-minamimachi, Chuo-ku, 650-0047 Kobe, Japan. [2] Laboratory for Physical Biology, RIKEN Center for Biosystems Dynamics Research (RIKEN BDR), 2-2-3 Minatojima-minamimachi, Chuo-ku, 650-0047 Kobe, Japan. [3] European Molecular Biology Laboratory (EMBL) Barcelona, Dr. Aiguader, 88, 08003 Barcelona, Spain. Correspondence and requests for materials should be addressed to M.E. (email: miki.ebisuya@riken.jp)

One of the goals of synthetic biology is creating a synthetic tissue to understand natural developmental mechanisms[1–3], to explore the origin of multicellularity[4] and to engineer a programmable tissue for therapeutic purposes[5,6]. The first step towards a synthetic tissue is controlling pattern formation, which enables to place different types of cells properly in a tissue. Several synthetic cellular patterns have been reported previously: Ring patterns were created in genetically engineered bacteria that can sense the concentrations of small molecules[7,8]. In mammalian cells, 2D and 3D patterns were created based on engineered cell sorting mechanisms[9,10]. However, there is another pattern formation mechanism that has not been artificially created in mammalian cells despite its importance: the reaction-diffusion (RD) patterning system.

The concept of a self-organizing RD system was first proposed by Alan Turing as a chemical system of interacting and diffusible molecules giving rise to various stable patterns, such as spots and stripes[11–14]. Recent studies have suggested that RD system underlies a number of developmental patterning phenomena, including digit formation in the limb[15,16], pigmentation on the skin[17], the formation of hair follicles and feather buds on the skin[18,19], branching morphogenesis in the lung[20] and rugae formation in the palate[21]. In the field of synthetic biology, a regular stripe pattern has been created in colonies of engineered bacteria, in which diffusion of the small molecule AHL regulates the motility of the actively swimming bacteria[22], and this patterning mechanism can be considered as a non-classic RD system. Very recently, a stochastic Turing pattern has been created in engineered bacteria that have a synthetic network of two diffusible small molecules[23]. However, an RD pattern has not so far been created in eukaryotic cells. Furthermore, although RD patterning in embryonic development is mediated mostly by the interaction of diffusible protein ligands called morphogens rather than by small molecules or cell movement[15,16,18–21], a protein-based RD patterning system has not so far been created either.

Our goal here thus was to engineer an RD patterning network of protein ligands in mammalian cells. An RD pattern requires a minimum of two diffusible molecules, or signaling pathways, and we chose to employ the well-studied Nodal–Lefty signaling pathway, which regulates mesodermal induction, axis formation and left-right patterning[24–26]. The Nodal–Lefty pathway has been proposed to meet two conditions for a stable RD pattern:[14,27] firstly, binding of Nodal to its receptor activates the production of both Nodal and Lefty whereas Lefty inhibits the activity of Nodal[24,28,29]. Secondly, the diffusion of Nodal is reported to be slower than Lefty in zebrafish, chick and mouse embryos[27–31]. In other words, Nodal and Lefty may act as a short-range activator and a long-range inhibitor, respectively, and thus satisfy the requirement for a classic Turing pattern proposed by Gierer and Meinherdt[12,13]. It remains undemonstrated, however, whether the Nodal–Lefty signaling can actually produce an RD pattern, as well as how Nodal and Lefty show different diffusivity.

In this study, we reconstitute an activator-inhibitor circuit of Nodal and Lefty to test if it leads to any pattern formation in mammalian cell culture. We also take advantage of the simple in vitro system and investigated the differences in the diffusion mechanisms of Nodal and Lefty.

## Results

**Pattern formation with an activator–inhibitor circuit**. We first created an activator circuit in HEK293 cells, where the activator Nodal induces the expression of Nodal itself (Fig. 1a). Extracellular Nodal is known to bind to the co-receptor, Cryptic or Cripto, as well as to the heterodimeric receptors, Activin receptor types I and II. The activated receptors then activate Smads that

form a complex with the transcription factor FoxH1, leading to the transcription of downstream targets[24]. Since HEK293 cells lack some of these essential components to transduce the Nodal signaling[32], we introduced exogenous Cryptic and FoxH1 to the cells (Supplementary Fig. 1). The induction of gene expression in response to Nodal signaling was monitored with an $(f2)_7$-luc reporter, a seven-times repeat of a FoxH1-responsive element that regulates the luciferase expression[33]. The exogenous expression of a type II receptor, Acvr2b, further improved the induction rate of the $(f2)_7$-luc reporter signal (Supplementary Fig. 1b). Co-culturing with Nodal-producing cells, instead of the recombinant Nodal proteins, also activated the $(f2)_7$-luc reporter cells (Supplementary Fig. 2a), showing that secreted Nodal propagates to the neighboring cells. Finally, we added the $(f2)_7$-Nodal construct to let the cells both produce and respond to Nodal, completing the positive feedback of the activator circuit (Fig. 1a). When the HEK293 cells engineered with the activator circuit were seeded at a high density (near confluent), the $(f2)_7$-luc reporter signal was initially detected only in ~10% of the cells, and those reporter-positive cells were randomly distributed (Fig. 1b, 0 h). Then the reporter-positive cells activated the immediate neighboring cells, and the small domains of the positive cells appeared at around 18 h. By 42 h, all the cells became reporter-positive (Fig. 1b, c; Supplementary Movie 1). Note that the cells are nearly confluent from the time zero and that the cell proliferation rate is low (the cells divided approximately twice in 70 h), meaning that the observed signal propagation is not because of the cell proliferation but because of the mutual activation of Nodal signaling among the neighboring cells. The propagated reporter signal lasted until 50–60 h and started to gradually decline possibly due to the lack of fresh medium and/or luciferin supply.

This activator circuit serves as a base for an activator-inhibitor circuit, where Nodal induces the expression of Nodal as well as that of Lefty2 that inhibits the Nodal signaling (Fig. 1d). We first confirmed that co-culturing with Lefty2-producing cells inhibits the activation of the $(f2)_7$-luc reporter cells (Supplementary Fig. 2b). Then we introduced the $(f2)_7$-Lefty2 construct into the HEK293 cells already engineered with the activator circuit, adding the negative feedback by Lefty (Fig. 1d). The activator–inhibitor circuit initially behaved similarly to the activator circuit: the small domains of reporter-positive cells appeared at around 18 h and became bigger (Fig. 1e). Then the domain growth slowed down at around 30 h, and a pattern of clear positive domains and negative domains was formed by 36 h (Fig. 1e, f; Supplementary Movie 1). Note that the reporter-positive cells and negative cells are genetically identical since the cell line was cloned. The pattern was reproducible (Fig. 1g), even after re-cloning of the cells (Fig. 1h), and the average size of positive domains was 196 ± 15 μm (Fig. 1i; Supplementary Fig. 3). The pattern did not change much after 36 h and kept essentially constant until 60 h (Fig. 1i, j), even though the entire signal started to decline at 50–60 h as observed with the activator circuit.

To assess the periodicity of our synthetic pattern, we calculated the spatial correlation of the image (Supplementary Fig. 4a–f). The second peak of the radially averaged correlation function indicates the distance over which the pattern repeats itself (i.e., the distance from peak-to-peak of a pattern). While the correlation of the image of the cells with the activator circuit rapidly dropped (Supplementary Fig. 4b), that with the activator–inhibitor circuit showed a small second peak at around 400 μm (Supplementary Fig. 4d), suggesting a weak periodicity of the pattern. This period was consistent with the sum of the average positive domain width and negative domain width (376 ± 99 μm) (Supplementary Fig. 4g). Although the image of the activator circuit in the middle of the signal propagation process

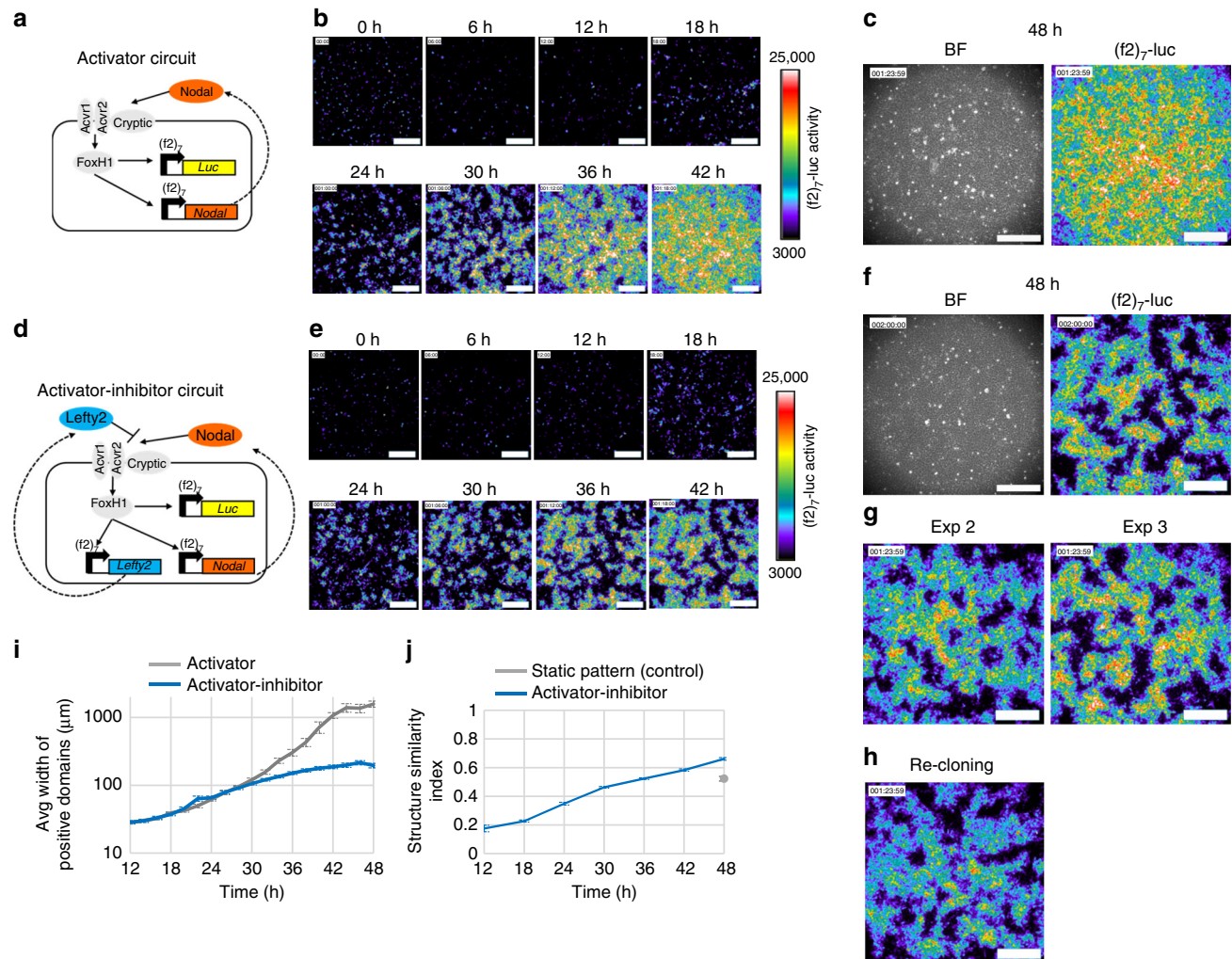

**Fig. 1** Cells with an activator–inhibitor circuit spontaneously give rise to a pattern. **a** The activator circuit. **b** Time-lapse imaging of the HEK293 cells engineered with the activator circuit. See also Supplementary Movie 1. **c** The bright field and luciferase images of the cells with the activator circuit at 48 h. **d** The activator–inhibitor circuit. **e** Time-lapse imaging of the HEK293 cells engineered with the activator-inhibitor circuit. See also Supplementary Movie 1. **f** The bright field and luciferase images of the cells with the activator–inhibitor circuit at 48 h. **g** Repeated experiments of **f**. **h** The luciferase image of an activator-inhibitor cell line at 48 h that was re-cloned from the cells shown in **f**. **i** The width of positive domains was measured at each time point as described in Supplementary Fig. 3. **j** The structural similarity (SSIM) index between two images at time $t$ and $t + 6$ h was calculated as described in Supplementary Fig. 3. A higher index means a more stable pattern. The gray dot at 48 h indicates the SSIM index of a control static sample, where the cells that constitutively express luciferase were mixed with wild-type cells. Scale bars: 400 μm (**b**, **c**, **e**–**h**). Source data are provided as a Source Data file (**i**, **j**)

showed positive domains and negative domains (Fig. 1b, 30 h), its correlation did not show a clear second peak (Supplementary Fig. 4f; Activator-inhibitor 48 h vs. Activator 30 h, $p = 0.029$, Wilcoxon rank sum test). These results indicate that our activator–inhibitor circuit, a reconstituted network of the Nodal–Lefty signaling, has an ability to make cells form a stable RD pattern with a periodicity of ~400 μm.

**Different distribution ranges of Nodal and Lefty**. Why can the synthetic Nodal–Lefty circuit give rise to a pattern? Since a stable RD pattern typically requires a short-range activator and a long-range inhibitor[11–14], the difference in the diffusion ranges of two diffusible molecules is critical. Although the diffusion of Nodal has been reported to be slower than that of Lefty in zebrafish, chick and mouse embryos[27–31], our experimental conditions are different from those of the previous studies especially because we culture the cells on a dish as a monolayer with plenty of culture medium. We thus tested whether Nodal and Lefty show different

diffusion ranges in our system. To visualize the distribution of Nodal and Lefty, we placed the ligand-producing cells and the receptor cells (i.e., wild-type cells) separately in adjacent areas by using a mold called a culture insert (Fig. 2a). Then the extra-cellular Nodal and Lefty were exclusively visualized with a split luciferase system called HiBiT (Fig. 2b): the N-terminal bigger half of NanoLuc, Large BiT (LgBiT), is added to the medium but does not enter a cell. Thus, the C-terminal smaller half of NanoLuc, the HiBiT tag, binds to LgBiT to reconstitute a func-tional luciferase only outside the cell (Fig. 2b). Avoiding intra-cellular signal this way is crucial since the concentrations of Nodal and Lefty are much higher inside the cell, which easily masks their extracellular distributions. Because Nodal and Lefty are cleaved by proteases to become their mature forms[28,34], we inserted the HiBiT tag into the middle part of the proteins, at the N-terminus of the mature domains (Fig. 2c, d).

The luminescence signal of HiBiT-Nodal displayed an extremely narrow distribution (Fig. 2c). The Nodal distribution reached at its equilibrium by four hours after the addition of

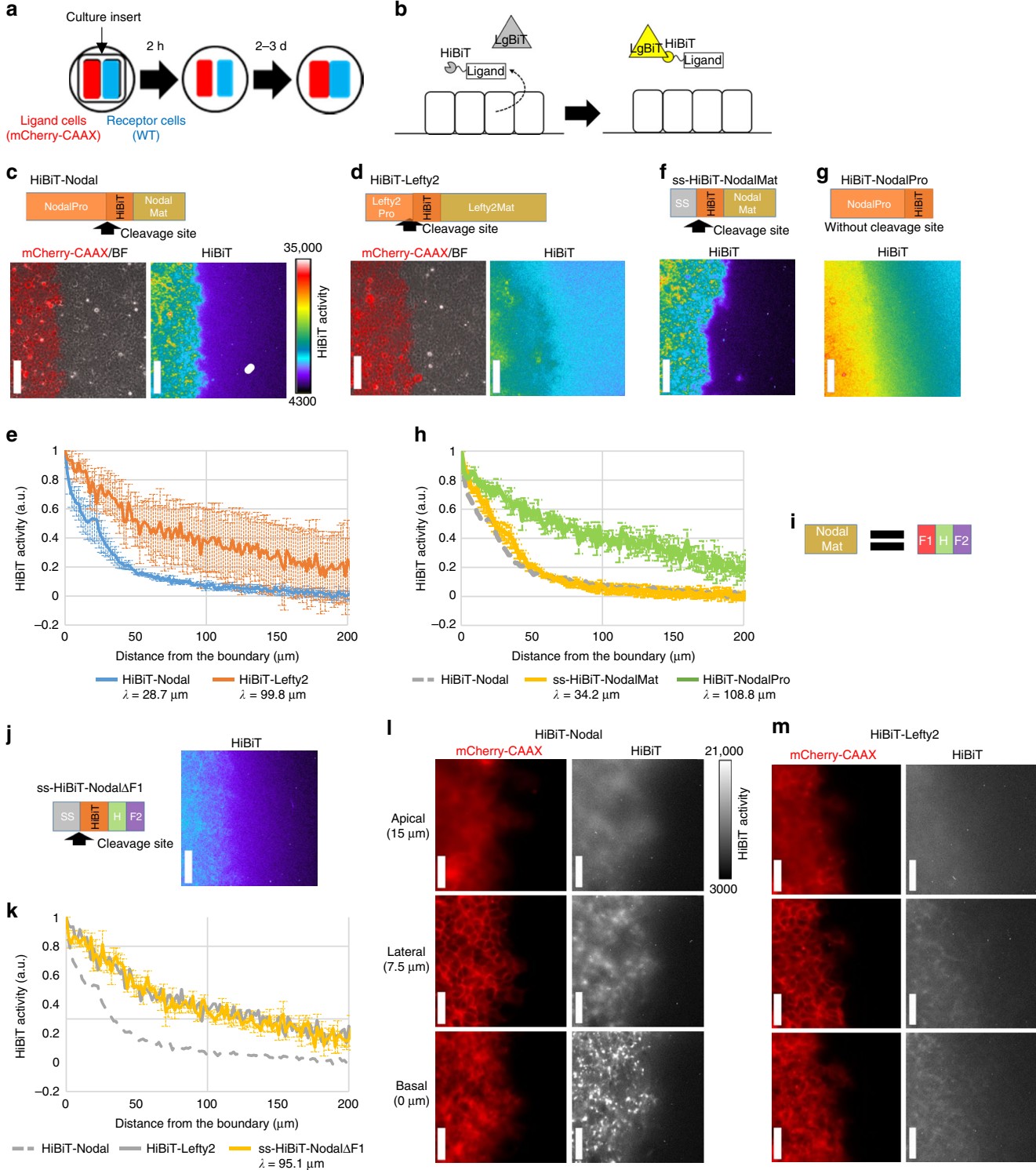

LgBiT and substrates (Supplementary Fig. 5a). To compare the distribution ranges of different proteins, we fitted the normalized distribution to a simple exponential function, $\exp(-x/\lambda)$. The characteristic distance $\lambda$ is the point where the signal drops to $1/e$, and $\lambda$ also represents $\sqrt{D/\gamma}$, where $D$ is the effective diffusion coefficient and $\gamma$ is the degradation rate[35,36]. In the case of HiBiT-Nodal, $\lambda = 28.7\ \mu m$ (Fig. 2e), suggesting that the effective range of Nodal is only one or two cells since the cell size is 10–20 μm. By contrast, HiBiT-Lefty2 displayed much wider distribution where the signal gradually decreased (Fig. 2d), and $\lambda = 99.8\ \mu m$ (Fig. 2e),

indicating that the effective range of Nodal is 3.5 times narrower than that of Lefty2.

The molecular sizes of Nodal and Lefty2 are similar (full-length Nodal: 354 aa; full-length Lefty2: 368 aa; mature Nodal: 110 aa; mature Lefty2: 291 aa) and thus unlikely to be the cause of their different diffusion ranges. We hypothesized the existence of a trap mechanism to confine extracellular Nodal in immediate neighboring cells, and focused on Nodal rather than Lefty. Since the full-length Nodal protein comprises the mature domain and prodomain (Fig. 2c), we examined which domain is responsible

**Fig. 2** The distribution range of Nodal is shorter than that of Lefty. **a** Culture insert assay. Ligand-producing cells (labeled with mCherry-CAAX) and receptor cells (wild-type cells) are cultured separately in a culture insert. After removal of the culture insert, the two cell types fill the cell-free gap, establishing a straight boundary. **b** HiBiT system to visualize the extracellular distribution of ligands. The small tag, HiBiT, is fused to the ligand, whereas the LgBiT and substrate are added to the medium. The HiBiT and LgBiT reconstitute NanoLuc only outside the cells. **c** Top: the HiBiT tag was inserted into the N-terminus of the Nodal mature domain. Bottom left: the boundary of the HiBiT-Nodal-producing cells (ligand cells, labeled with mCherry-CAAX) and receptor cells. The mCherry-CAAX image was merged with the bright field image. Bottom right: HiBiT-mediated luminescence image showing the extracellular distribution of HiBiT-Nodal. **d** Top: the HiBiT tag was inserted into the N-terminus of the Lefty2 mature domain. Bottom left: the boundary of the HiBiT-Lefty2-producing cells and receptor cells. Bottom right: the extracellular distribution of HiBiT-Lefty2. **e** Quantified distribution profiles of HiBiT-Nodal and HiBiT-Lefty2. Each distribution was fitted to exp(-x/λ) to estimate the characteristic distance λ. **f** Top: a signal sequence (ss) and the HiBiT tag were fused to the N-terminus of the Nodal mature domain. Bottom: the extracellular distribution of ss-HiBiT-NodalMat. **g** Top: the HiBiT tag was fused to the C-terminus of the Nodal prodomain. Bottom: the extracellular distribution of HiBiT-NodalPro. **h** Quantified distribution profiles of ss-HiBiT-NodalMat and HiBiT-NodalPro. The HiBiT-Nodal distribution shown in e is displayed as a control. **i** The Nodal mature domain consists of three subdomains: the finger 1 (F1), heel (H) and finger 2 (F2). **j** Left: the finger 1 domain was deleted from ss-HiBiT-NodalMat. Right: the extracellular distribution of ss-HiBiT-NodalΔF1. **k** Quantified distribution profiles of ss-HiBiT-NodalΔF1. The distributions of HiBiT-Nodal and HiBiT-Lefty2 shown in e are displayed as a control. **l, m** Higher magnification view of HiBiT-Nodal (**l**) and HiBiT-Lefty2 (**m**). The mCherry-CAAX images were normalized differently between **l** and **m**. Data are means and s.e.m. (*n* = 3) (**e**, **h**, **k**). Scale bars: 200 μm (**c**, **d**, **f**, **g**, **j**); 50 μm (**l**, **m**). Source data are provided as a Source Data file (**e**, **h**, **k**)

for the narrow distribution. Whereas the mature domain of Nodal displayed a narrow distribution just like the full-length Nodal (Fig. 2f), the prodomain displayed a wider distribution just like Lefty2 (Fig. 2g, h). The Nodal mature domain further comprises three subdomains[37]: the finger 1 domain, heel domain and finger 2 domain (Fig. 2i). Deleting the finger 1 domain from the Nodal mature protein made the distribution wider (Fig. 2j, k), indicating that the finger 1 subdomain of Nodal is responsible for its narrow distribution.

**Extracellular Nodal localizes underneath the cells**. We further investigated how the finger 1 domain limits the distribution range of Nodal. While a previous study has suggested that binding of Nodal to the Acvr2b receptor slows down the Nodal diffusion in zebrafish[38], the overexpression or deletion of *Acvr2b* in our system did not change the Nodal distribution range (Supplementary Fig. 5b, c). Then we checked the localization of extracellular Nodal, noticing that the HiBiT-Nodal signal was in focus at the basal side of cells but out of focus at the lateral or apical side (Fig. 2l). The basal side was judged with the dense structure of cell membrane, and the lateral and apical sides were defined as the points 7.5 and 15 μm above the basal side, respectively (Fig. 2l, m; for higher resolution images, see Supplementary Fig. 6). Consistent with this observation, extracellular Nodal is suggested to localize underneath the cells even in mouse embryos[39]. We also noticed that the HiBiT-Nodal near the basal side formed small clusters (Fig. 2l). By contrast, the HiBiT-Lefty2 signal was blurry both at the basal and apical sides (Fig. 2m). These results suggest that extracellular Nodal is confined in the space between the cells and the culture dish as clusters, which may be the cause of the narrow distribution of Nodal.

**Mathematical models of the pattern forming circuit**. To understand the patterning mechanism of our activator–inhibitor circuit in more detail, we constructed simple mathematical RD models (Fig. 3a–j; Methods). Two mechanisms have been reported regarding how Lefty inhibits the Nodal signaling: Lefty competes with Nodal for the co-receptor and receptors[28,40] (Fig. 3a), or Lefty directly binds to and then inhibits Nodal[40] (Fig. 3e). We thus constructed two types of model by using parameters we measured or estimated (Supplementary Fig. 7): the "competitive inhibition model" and the "competitive inhibition + direct inhibition model". Both models gave rise to patterns comprising positive domains and negative domains when the parameters were in the right ranges (Fig. 3b, f). The patterns resulting from the competitive inhibition model were periodic

(Fig. 3d), and the patterning parameter range was almost identical with the parameter range that satisfied Turing instability[13] (compare Fig. 3c with 3b), the condition for Turing pattern formation, meaning that these patterns are classic Turing patterns. However, Turing patterns are not the only type of RD system that can perform spatial patterning. The competitive inhibition + direct inhibition model also gave rise to patterns, when the strength of direct inhibition was in the right ranges (Fig. 3h, i; Supplementary Fig. 8). The domains resulting from the competitive inhibition + direct inhibition model showed less regular size and shape (Fig. 3h, i) compared with those of the Turing pattern, and the Turing instability condition was not satisfied in all the parameter regions tested with this model (Fig. 3g). This non-Turing patterning mechanism is essentially the same as the formation of "solitary localized structures"[41,42] caused by an excitable or bistable system combined with a rapidly diffusing inhibitor: the positive domains are formed by short-range self-activation initially, and the propagation of domains are stopped by long-range inhibition in the later stage. Thus, we named the less-regular patterns as "solitary patterns". One qualitative difference between the solitary and Turing patterns is that a solitary pattern depends on the initial condition while a Turing pattern does not essentially vary according to the initial condition (Supplementary Fig. 9). The actual cell pattern resulting from our activator–inhibitor circuit varied when we altered the initial condition by mixing reporter-positive cells and negative cells at different ratios (Supplementary Fig. 9), suggesting that our synthetic RD pattern may be a solitary pattern rather than a Turing pattern.

An important step in theoretical modeling is to alter parameters in the model, and then test the observed predictions experimentally. We thus attempted to alter the maximum synthesis rate of the activator-inhibitor circuit both in simulation and living cells. Our model predicted that increasing the maximum synthesis rate of Nodal or Lefty should change the ratio of the positive domains to negative domains in the resulting patterns (Fig. 3h–j). To experimentally increase the maximum synthesis rates, we introduced the extra copies of (f2)$_7$-*Lefty2* or (f2)$_7$-*Nodal* to the cells already engineered with the activator–inhibitor circuit (Fig. 3k, l). The constitutive expression of *PGK-mCherry* or *PGK-GFP* from the same construct as (f2)$_7$-*Lefty2* or (f2)$_7$-*Nodal* was used as a marker of the increased copy numbers (Fig. 3m). As predicted, increasing the maximum synthesis rate (i.e., the copy number) of Nodal made the cells homogeneously positive, whereas that of Lefty expanded the area of negative domains (compare Fig. 3n with Fig. 3h–j).

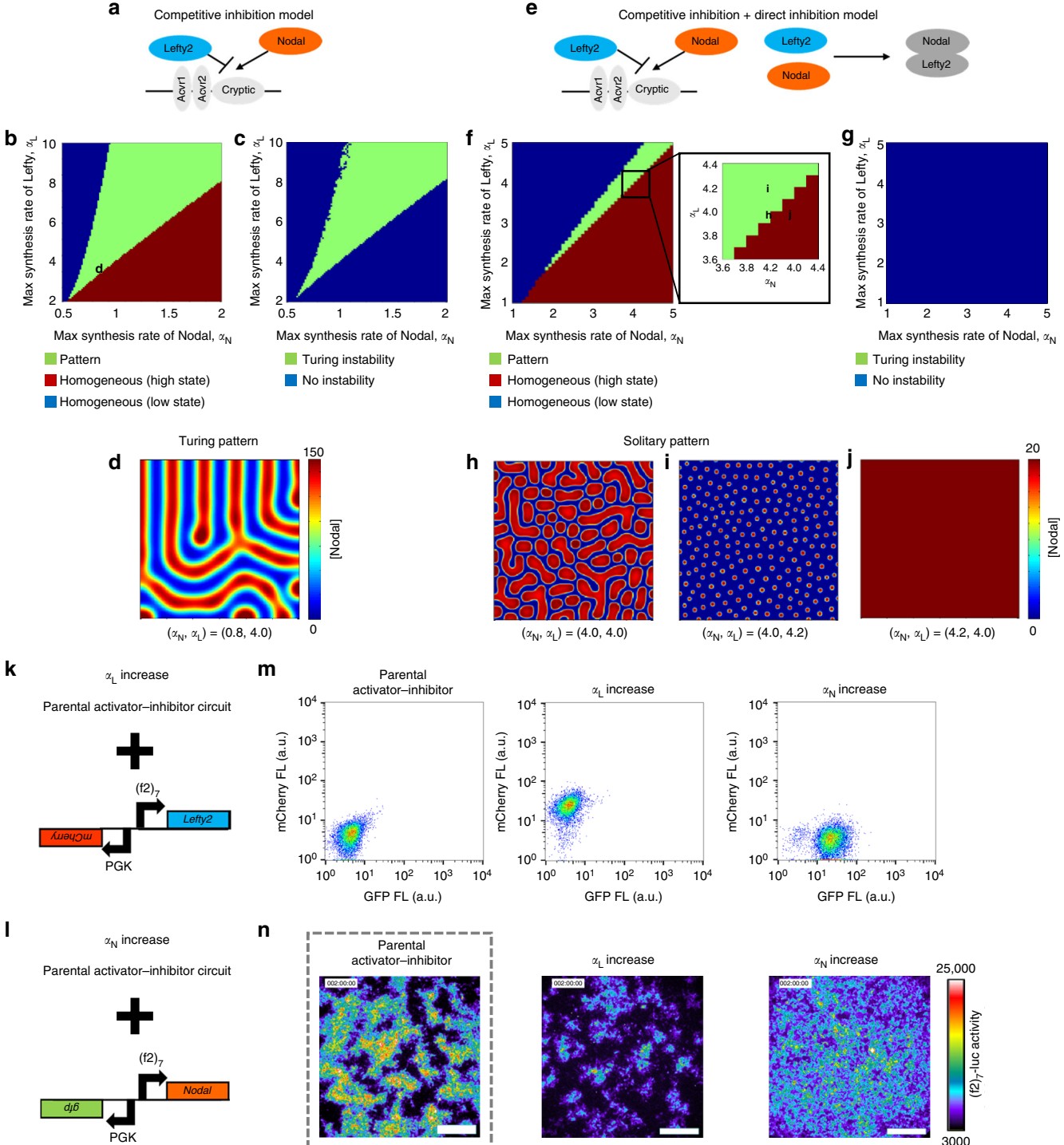

**Fig. 3** Mathematical models of the activator-inhibitor circuit. **a–d** The competitive inhibition model. **a** A scheme of competitive inhibition. **b** The parameter region for pattern formation (green). The competitive inhibition model was simulated in one dimension. See also Methods. The parameter combination used in **d** is indicated by the character d. **c** The parameter region that meets the Turing instability condition (green). **d** Two-dimensional simulation with $(\alpha_N, \alpha_L) = (0.8, 4.0)$. A typical Turing pattern was formed with this model and parameter set. **e–j** The competitive inhibition + direct inhibition model. **e** A scheme of competitive inhibition and direct inhibition. **f** The parameter region for pattern formation (green). The competitive inhibition + direct inhibition model was simulated in one dimension. The parameter combinations used in **h–j** are indicated in the inset. **g** The Turing instability condition was not satisfied in the entire parameter regions tested. **h–j** Two-dimensional simulation with different combinations of the parameters $\alpha_N$ and $\alpha_L$. Solitary patterns, not Turing patterns, were formed with this model and parameter range. **k** To increase $\alpha_L$, a construct containing $(f2)_7$-Lefty2 and PGK-mCherry was added to the parental activator–inhibitor circuit shown in Fig. 1d. **l** To increase $\alpha_N$, a construct containing $(f2)_7$-Nodal and PGK-GFP was added to the parental activator–inhibitor circuit. **m** FACS plots confirming the introduction of the additional constructs. The mCherry signal indicates extra copies of $(f2)_7$-Lefty2 ($\alpha_L$ increase), and the GFP signal indicates extra copies of $(f2)_7$-Nodal ($\alpha_N$ increase). **n** Patterns resulting from the activator–inhibitor circuit with the increased $\alpha_L$ or $\alpha_N$. The pattern of the parental activator-inhibitor cells shown in Fig. 1f is displayed as a control. Scale bars: 400 μm

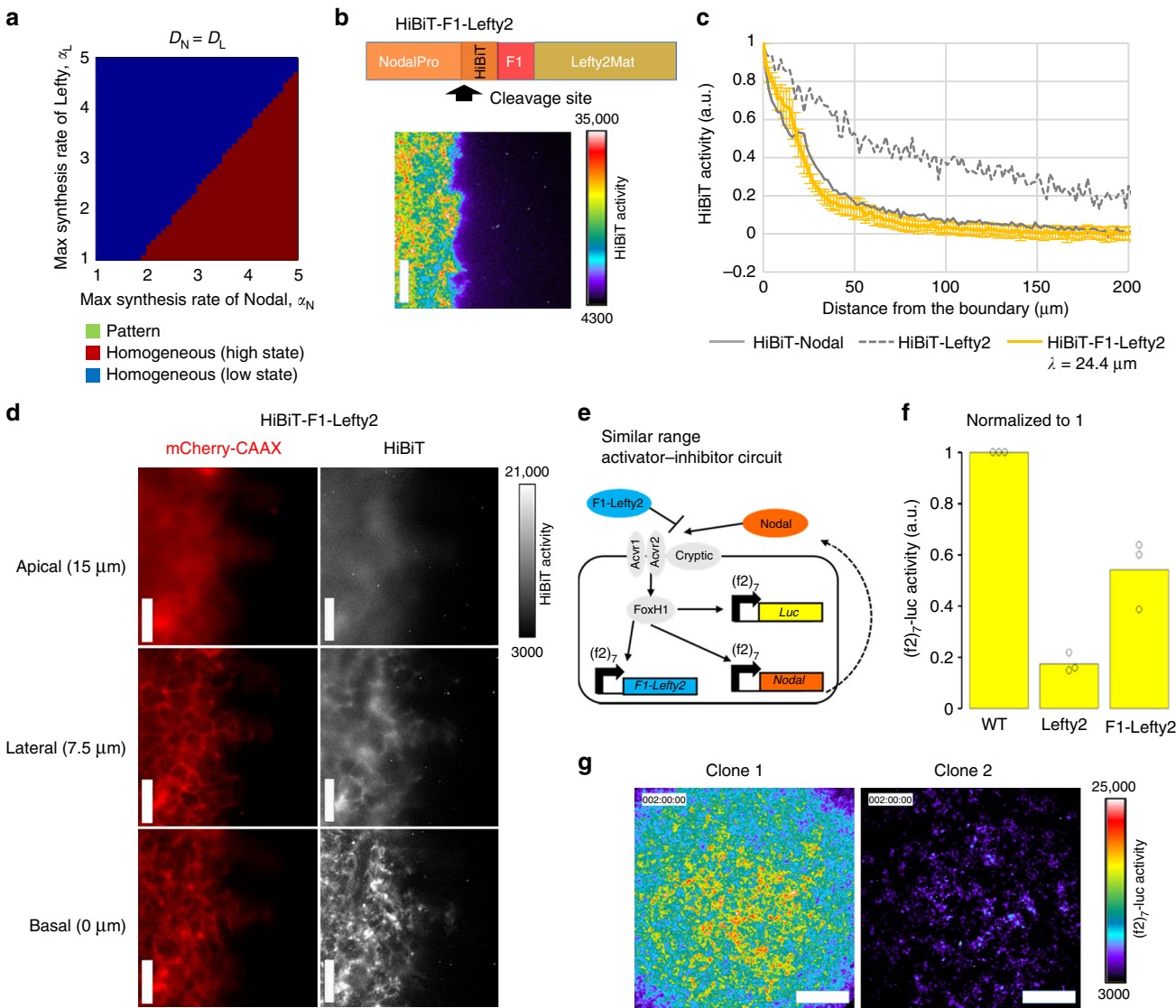

**Fig. 4** Different diffusivity of Nodal and Lefty is crucial for the pattern formation. **a** The parameter region for pattern formation does not exist with the same diffusion coefficients for Nodal and Lefty ($D_N = D_L = 1.96$ $\mu m^2$ $min^{-1}$). The competitive inhibition + direct inhibition model was simulated in one dimension. **b** Top: the finger 1 domain of Nodal was fused to Lefty2. Bottom: the culture insert assay showing the extracellular distribution of HiBiT-F1-Lefty2. **c** Quantified distribution profile of HiBiT-F1-Lefty2. The distributions of HiBiT-Nodal and HiBiT-Lefty2 shown in Fig. 2e are displayed as a control. Data are means and s.e.m. ($n = 3$). **d** Higher magnification view of HiBiT-F1-Lefty2. **e** The similar range activator-inhibitor circuit. F1-Lefty2 was used instead of Lefty2 in the activator-inhibitor circuit to make the diffusion coefficients of Nodal and Lefty similar. **f** Inhibitory activity of F1-Lefty2. Wild-type cells, Lefty2-producing cells or F1-Lefty2-producing cells were co-cultured with the cells engineered with the activator circuit, and the $(f2)_7$-luc activities were measured. Data are means and individual points ($n = 3$). **g** The image of two representative clones of the cells engineered with the similar range activator–inhibitor circuit at 48 h. Scale bars: 200 $\mu m$ (**b**); 50 $\mu m$ (**d**); 400 $\mu m$ (**g**). Source data are provided as a Source Data file (**c**, **f**)

**Manipulating the diffusion coefficient of Lefty**. We further altered another important parameter, the diffusion coefficient of the activator–inhibitor circuit. Our model predicted that no pattern should be formed, irrespective of a Turing pattern or a solitary pattern, if the diffusion coefficients of Nodal and Lefty are the same (compare Fig. 4a with Fig. 3f). To experimentally test this prediction, we fused the finger 1 domain of Nodal to Lefty2 (Fig. 4b). As expected, the HiBiT-F1-Lefty2 displayed a narrow distribution just like Nodal (Fig. 4b, c). The HiBiT-F1-Lefty2 also localized near the basal side of cells and formed small clusters, just like Nodal (compare Fig. 4d with Fig. 2l, m). These results show that the Nodal finger 1 domain is indeed important for the narrow distribution and able to make the distribution of Lefty2 narrow upon transplantation.

Then we created an activator-inhibitor circuit with *F1-Lefty2*, instead of wild-type *Lefty2*, and named the new circuit as a "similar range activator–inhibitor circuit" (Fig. 4e). F1-Lefty2 inhibited the Nodal signaling although its inhibitory activity was a little weaker than that of wild-type Lefty2 (Fig. 4f). When the $(f2)_7$-*F1-Lefty2* construct was added to the activator circuit, the engineered cells did not show a pattern but displayed an almost homogeneous image (Fig. 4g), and the spatial correlation dropped rapidly without showing a second peak (Supplementary Fig. 10). These results confirmed our prediction that different diffusion ranges of Nodal and Lefty are crucial for the pattern formation through our activator–inhibitor circuit.

## Discussion

We have created here, to our knowledge, the first synthetic RD pattern in mammalian cells, which is verified by comparison of various experimental perturbations with a theoretical model (i.e., changing the diffusion constants and maximum expression levels). In doing so, we show that the pattern formation through our synthetic circuit is driven by the different diffusion ranges of Nodal and Lefty, which are influenced by the Nodal finger 1 domain and the confinement of Nodal underneath the cells.

Extracellular Nodal localized underneath the basal side of cells. Although the mechanism for this localization remains unclear, one possibility is that Nodal is trapped by protein complexes that exist between the cells and a dish, such as the extracellular matrix (ECM) and adhesion complex. In fact, Nodal is reported to interact with sulfated glycosaminoglycans that localize to the basement-membrane like structure underneath the cells in mouse embryos[39]. Since Nodal and Lefty are known to display different diffusivity in developing mouse, chick and zebrafish embryos, it will be interesting to examine whether the Nodal localization as well as the finger 1 domain affect the distribution range of Nodal in those embryos. The localization underneath the cells enabled Nodal to form a steep gradient in our cell culture even with plenty of culture medium. Without a proper trapping mechanism, the free diffusion in the medium should be too fast for Nodal to form any distribution. Recent studies show the gradient distribution formation of other morphogens in cell culture[43], suggesting that localization underneath the cells may be a common trapping mechanism.

Nodal displayed a 3.5 times shorter distribution range than Lefty2 in our measurements. If the degradation rates are similar between Nodal and Lefty2, the effective diffusion coefficient of Nodal should be 12 times smaller than that of Lefty2. According to our measurements, the degradation of Lefty2 is actually 2.4 times faster than that of Nodal (Supplementary Fig. 7b), suggesting that the effective diffusion coefficient of Nodal is 29 times smaller than that of Lefty2, which is comparable to the value reported in zebrafish[27]. Direct measurements of the diffusion coefficients will be necessary to verify these numbers even though our attempts for FRAP analysis were not successful due to too weak signal of fluorescent fusion Nodal and Lefty. In any case, a sufficiently large difference in the diffusivity of Nodal and Lefty was proven critical for our pattern formation. Very recently, the differential diffusivity of Nodal and Lefty has also been reported to underlie scaling of the proportions of germ layers in zebrafish[44].

The cells engineered with our activator–inhibitor circuit gave rise to a pattern, which we believe is the first mammalian example of a synthetic RD pattern. Our mathematical models suggested two possible RD systems: a Turing pattern and a solitary pattern. Both patterns require the positive feedback of a short-range activator and the negative feedback of a long-range inhibitor. The uniform stationary state is absolutely unstable in Turing periodic patterns, whereas it is stable in solitary spot patterns. Therefore, a solitary pattern can be initiated at the position where sufficiently strong local perturbation is applied. By contrast, a Turing pattern can appear without any significant perturbation. Although the fact that our cell pattern significantly varied according to the initial condition favors a solitary pattern over a Turing pattern, further experiments are needed to distinguish the two possibilities, including the direct measurement of the association rate of Nodal and Lefty in the competitive inhibition + direct inhibition model.

While pattern formation is fundamental for embryonic development, investigating the underlying molecular mechanisms in complex living tissues is often difficult. A simple synthetic system in cell culture should offer a unique opportunity to investigate the

mechanisms of pattern formation and morphogen diffusion in detail. This work will also serve as a base for engineering a more complex synthetic tissue[1,3–6].

## Methods

**DNA constructs.** The genetic constructs used in this study are listed in Supplementary Table 1. The promoters or genes were subcloned into pDONR vector to create entry clones. These entry clones were recombined with *piggyBac* vector (a gift from Knut Woltjen)[45] or Tol2 vector (a gift from Koichi Kawakami)[46,47] by using the Multisite Gateway technology (Invitrogen). The (f2)₇ promoter was created by fusing the (f2)₇ enhancer sequence (a gift from Hiroshi Hamada)[33] to the CMV minimal promoter. The CAG promoter is a gift from Junichi Miyazaki[48]. Genes related to Nodal signaling (*Nodal*, *Lefty2*, *Cryptic*, *FoxH1* and *Acvr2b*) were cloned from mouse cDNA Mix (GenoStaff). CAAX domain was cloned from *Kras4B*. The CRISPR guide sequences for *Acvr2b* deletion were cloned into pSpCas9(BB)-2A-Puro vector (a gift from Feng Zhang, addgene #62988)[49].

**Cell culture.** 293AD (Cell Biolabs), a cell line derived from parental HEK293 cells, was used for all experiments because of its flattened morphology and firm attachment to a culture dish. The cells were maintained in DMEM/F12 medium containing 10% fetal bovine serum at 37 °C with 5% $CO_2$.

**Creation of stable cell clones.** The genetic constructs were introduced into HEK293 cells with the *piggyBac* or Tol2 transposase. To create the reporter cell line for Nodal signaling, (f2)₇-luc, CAG-Cryptic, CAG-FoxH1 and CAG-Acvr2b were introduced into HEK293 cells. To create the activator circuit, (f2)₇-Nodal was added to the reporter cell line. To create the activator–inhibitor circuit, (f2)₇-Lefty2 was added to the activator cell line. To increase the maximum synthesis rate of Nodal or Lefty, (f2)₇-Nodal with PGK-GFP or (f2)₇-Lefty2 with PGK-mCherry was further added to the activator-inhibitor cell line. To create the similar range activator-inhibitor circuit, (f2)₇-F1-Lefty2 was added to the activator cell line instead of (f2)₇-Lefty2. To create HiBiT-tagged ligand cell lines, CAG-ligand and EF1a-mCherry-CAAX were introduced into HEK293 cells. After antibiotics selection, all cell lines except for the cells used in Fig. 4f and Supplementary Figs. 1b and 7b were cloned from a single cell. As for the activator circuit, 5 out of 6 clonal lines analyzed showed successful signal propagation, and 1 line was used as a base for the activator-inhibitor circuit. As for the activator-inhibitor circuit, 2 out of 9 clonal lines analyzed showed successful cell patterning, and 1 line was rigorously characterized. As for the similar range activator–inhibitor circuit, 8 out of 12 clonal lines analyzed showed bright homogeneous images (represented by clone 1 in Fig. 4g), and 4 lines showed dark homogeneous images (represented by clone 2).

**Time-lapse imaging of synthetic pattern formation.** The $2.0 \times 10^5$ cells (200 μl suspension) were seeded onto the glass part (φ12 mm) of a glass base 35 mm dish (IWAKI). After 2 h incubation, the medium was replaced with the 2 ml fresh medium containing 20 mM HEPES and 100 μM D-luciferin, and the luminescence was imaged at each time point with a customized incubator microscope LCV110 (Olympus).

**Culture insert assay with the HiBiT system.** A culture insert (Ibidi) was placed on the glass part of a glass base 35 mm dish, and then the ligand cells and receptor cells ($4 \times 10^4$ cells each) were seeded separately in the two wells of the culture insert. After 2 h incubation, the culture insert was removed, and the cells were cultured with 2 ml medium. After 2–3 days incubation, the medium was replaced with the 2 ml fresh medium containing 20 mM HEPES, 1 μl HiBiT substrate (Nano-Glo® Live Cell EX-4377, Promega) and 8 μl LgBiT (Promega), and the luminescence and fluorescence were imaged with LCV110. Cells labeled with mCherry-CAAX were imaged with a confocal microscope LSM 780 (Carl Zeiss) in Supplementary Fig. 6.

**Quantification of HiBiT activity.** A $30 \times 300$ pixels ($48 \times 480$ μm²) rectangular area was set so that the short side of the rectangle is in parallel with the boundary between the ligand cells and receptor cells. The HiBiT activities and mCherry intensities were averaged along the short side. The averaged mCherry intensities were normalized with the following function:

$$N_{\text{Che}}(x) = \frac{A_{\text{Che}}(x) - \min(A_{\text{Che}}(x), x \in [1, 300])}{\max(A_{\text{Che}}(x), x \in [1, 300]) - \min(A_{\text{Che}}(x), x \in [1, 300])} \quad (1)$$

where $A_{\text{Che}}(x)$ is the averaged mCherry intensity at position $x$ pixel ($x = 1$ is the in the region of ligand cells and $x = 300$ is in the region of receptor cells). The averaged HiBiT activities were normalized with the following function:

$$N_{\text{HiBiT}}(x) = \frac{A_{\text{HiBiT}}(x)}{\frac{1}{50} \sum_{x=x_{0.5}-50}^{x_{0.5}-1} A_{\text{HiBiT}}(x)} \quad (2)$$

where $A_{\text{HiBiT}}(x)$ is the averaged HiBiT activity at position $x$, and $x_{0.5}$ is the position where the normalized mCherry intensity drops to 0.5. The normalized HiBiT

activities of three independent experiments were averaged and then fitted to the following function:

$$(1 - C)e^{-x/\lambda} + C \tag{3}$$

where $C < 1$ is background. Finally, the HiBiT activity distributions were given by

$$I_{HiBiT}(x) = \frac{N_{HiBiT}(x + x_{0.5}) - C}{1 - C} \tag{4}$$

The distance was compensated by 1 pixel = 1.6 μm.

**Luciferase assay.** For the luciferase assay shown in Supplementary Fig. 1b, the cells were seeded in a 24-well plate at $1.0 \times 10^5$ cells/well. After 24 h culture in the absence or presence of 10 nM recombinant Nodal, the cells were washed with PBS and eluted with 150 μl 1× lysate buffer (Luciferase assay system, Promega). For the intermingled co-culture assay (Fig. 4f; Supplementary Fig. 2), the ligand cells and reporter cells were seeded in a 24-well plate at $1.0 \times 10^5$ cells each/well and mixed. After 48 h co-culture, the cells were washed with PBS and eluted with 250 μl 1× lysate buffer. For the measurement of the signal response curve (Supplementary Fig. 7a), the reporter cells were seeded in a 96-well plate at 7000 cells/well. After 1 h incubation, the medium was changed into the fresh medium containing recombinant Nodal and Lefty1. After 48 h culture, the cells were washed with PBS and eluted with 100 μl 1× lysate buffer. The 20 μl lysate prepared above was mixed with 50 μl luciferase substrate (Luciferase assay system, Promega), and the luminescence was measured with a luminometer TriStar² (Berthold technologies).

**Degradation rate.** The $3 \times 10^5$ cells expressing each HA-tagged ligand were seeded in a 35 mm dish. After 2 days culture, the cells were treated with cycloheximide (50 μg/ml) and sampled for immunoblotting at 0, 1, 2, 3 and 6 h. Immunoblotting was performed according to a standard protocol, and the blot was probed with mouse anti-HA antibody (901501, Biolegend; 1/3000) and sheep anti-mouse antibody (NA931, GE Healthcare; 1/8000). The resulting bands were quantified with an ImageJ plug-in Gel Analyzer.

**FACS analysis.** The fluorescent cells were quantified with JSAN cell sorter (Bay bioscience) and analyzed with FlowJo software.

**Models.** Two simple mathematical RD models of the activator–inhibitor circuit were constructed. In the competitive inhibition model, Lefty was assumed to inhibit Nodal by competing for the co-receptor and receptors.

$$\frac{\partial N(\boldsymbol{x}, t)}{\partial t} = \alpha_N \frac{N^{n_N}}{N^{n_N} + \left[K_N\left\{1 + \left(\frac{L}{K_L}\right)^{n_L}\right\}\right]^{n_N}} - \gamma_N N + D_N \frac{\partial^2 N}{\partial \boldsymbol{x}^2} \tag{5}$$

$$\frac{\partial L(\boldsymbol{x}, t)}{\partial t} = \alpha_L \frac{N^{n_N}}{N^{n_N} + \left[K_N\left\{1 + \left(\frac{L}{K_L}\right)^{n_L}\right\}\right]^{n_N}} - \gamma_L L + D_L \frac{\partial^2 L}{\partial \boldsymbol{x}^2} \tag{6}$$

where $N$, $\alpha_N$, $n_N$, $K_N$, $\gamma_N$ and $D_N$ are the concentration, maximum synthesis rate, Hill coefficient, dissociation rate, degradation rate and diffusion coefficient of Nodal, respectively, and $L$, $\alpha_L$, $n_L$, $K_L$, $\gamma_L$, and $D_L$ are those of Lefty. In the competitive inhibition + direct inhibition model, Lefty was also assumed to inhibit

Nodal by directly binding to it.

$$\frac{\partial N(\boldsymbol{x}, t)}{\partial t} = \alpha_N \frac{N^{n_N}}{N^{n_N} + \left[K_N\left\{1 + \left(\frac{L}{K_L}\right)^{n_L}\right\}\right]^{n_N}} - \gamma_N N - k_+ NL + D_N \frac{\partial^2 N}{\partial \boldsymbol{x}^2} \tag{7}$$

$$\frac{\partial L(\boldsymbol{x}, t)}{\partial t} = \alpha_L \frac{N^{n_N}}{N^{n_N} + \left[K_N\left\{1 + \left(\frac{L}{K_L}\right)^{n_L}\right\}\right]^{n_N}} - \gamma_L L - k_+ NL + D_L \frac{\partial^2 L}{\partial \boldsymbol{x}^2} \tag{8}$$

where $k_+$ is the association rate of Nodal and Lefty.

Numerical simulations of the models were performed by using simple Euler method or ode45 solver of MATLAB (Mathworks). The diffusion terms were numerically solved by using the finite difference method. Parameters values shown in Table 1 were used unless stated otherwise ($k_+ = 0$ for the competitive inhibition model).

To create the phase diagram of pattern forming ability, 1D simulations with different combinations of $\alpha_N$ and $\alpha_L$ were performed for much longer time than the time scale of our experiment. As the initial state, two pulses of Nodal and Lefty concentrations were set. The resulting Nodal distribution was judged as a "pattern" when the maximum Nodal concentration was more than 2 times higher than the minimum Nodal concentration. Otherwise, the distribution was judged as a "high state" or "low state", depending on if the maximum Nodal concentration was higher than 0.01 or not.

The Turing instability condition was judged based on the following four inequalities:[13]

$$f_N + g_L < 0, \; f_N g_L - f_L g_N > 0, \\ d f_N + g_L > 0, \; (d f_N + g_L)^2 - 4d(f_N g_L - f_L g_N) > 0, \tag{9}$$

where $f_N$, $f_L$ are the derivatives of $f$ with respect to $N$, $L$, respectively, and $g_N$, $g_L$ are the derivatives of $g$ with respect to $N$, $L$. Functions $f$ and $g$ are given by

$$f = \alpha_N \frac{N^{n_N}}{N^{n_N} + \left[K_N\left\{1 + \left(\frac{L}{K_L}\right)^{n_L}\right\}\right]^{n_N}} - \gamma_N N \tag{10}$$

$$g = \alpha_L \frac{N^{n_N}}{N^{n_N} + \left[K_N\left\{1 + \left(\frac{L}{K_L}\right)^{n_L}\right\}\right]^{n_N}} - \gamma_L L \tag{11}$$

for the competitive inhibition model, or

$$f = \alpha_N \frac{N^{n_N}}{N^{n_N} + \left[K_N\left\{1 + \left(\frac{L}{K_L}\right)^{n_L}\right\}\right]^{n_N}} - \gamma_N N - k_+ NL \tag{12}$$

$$g = \alpha_L \frac{N^{n_N}}{N^{n_N} + \left[K_N\left\{1 + \left(\frac{L}{K_L}\right)^{n_L}\right\}\right]^{n_N}} - \gamma_L L - k_+ NL \tag{13}$$

for the competitive inhibition + direct inhibition model.

**Spatial correlation analysis.** The correlation function of the florescent intensity is given by

$$C(x, y) = \frac{\sum_{X,Y}(I(X, Y) - \bar{I})(I(X + x, Y + y) - \bar{I})}{\sum_{X,Y}(I(X, Y) - \bar{I})^2} \tag{14}$$

where the summation was taken over all pixel points $(X, Y)$ and $\bar{I}$ is the mean intensity. The radial correlation function $C(r)$ was calculated by averaging the

**Table 1 List of parameters for simulations**

| Parameter | Description | Value | Source |
|---|---|---|---|
| $\alpha_N$ | Maximum production rate of Nodal | 4.0 nM min⁻¹ | Arbitrarily chosen and varied |
| $\alpha_L$ | Maximum production rate of Lefty | 4.0 nM min⁻¹ | Arbitrarily chosen and varied |
| $k_+$ | Association rate of Nodal and Lefty | 0.03 min⁻¹ nM⁻¹ | Arbitrarily chosen and varied |
| $n_N$ | Hill coefficient of activation by Nodal | 2.63 | Supplementary Fig. 7a |
| $n_L$ | Hill coefficient of inhibition by Lefty | 1.09 | Supplementary Fig. 7a |
| $K_N$ | Dissociation coefficient of Nodal | 9.28 nM | Supplementary Fig. 7a |
| $K_L$ | Dissociation coefficient of Lefty | 14.96 nM | Supplementary Fig. 7a |
| $\gamma_N$ | Degradation rate of Nodal | $2.37 \times 10^{-3}$ min⁻¹ | Supplementary Fig. 7b |
| $\gamma_L$ | Degradation rate of Lefty | $5.65 \times 10^{-3}$ min⁻¹ | Supplementary Fig. 7b |
| $D_N$ | Diffusion coefficient of Nodal | 1.96 μm² min⁻¹ | Calculated from $\gamma_N$ and $\lambda$ for Nodal (measured in Fig. 2e) |
| $D_L$ | Diffusion coefficient of Lefty | 56.39 μm² min⁻¹ | Calculated from $\gamma_L$ and $\lambda$ for Lefty (measured in Fig. 2e) |

correlation function $C(X, Y)$ with the constraint $r = \sqrt{x^2 + y^2}$, which is formally given by

$$C(r) = \frac{1}{2\pi r} \iint C(x,y)\delta\left(r - \sqrt{x^2 + y^2}\right) dxdy \qquad (15)$$

where $\delta(x)$ is the Dirac's delta function.

**Code availability**. Scripts used in numerical simulations and analyses were written using MATLAB and are available upon request.

## Data availability

The authors declare that all the data supporting the results of this study are available within the article and its Supplementary Information files and from the corresponding authors upon reasonable request.

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

## Acknowledgements

We thank M. Matsuda for helping experiments, X. Diego and J. Sharpe for helping data analyses, H. Hamada for helpful advice and the members of Ebisuya lab for technical assistance and discussion. This work was supported by Precursory Research for Embryonic Science and Technology (PRESTO) (JPMJPR12A6 to M.E.), Grant-in-Aid for Scientific research (KAKENHI) programs from Ministry of Education Culture, Sports, Science, and Technology (MEXT) (16KT0080 to M.E., 26891027 to R.S.) and RIKEN Special Postdoctoral Researchers (SPDR) fellowship to R.S.

## Author contributions

R.S. and M.E. designed the work and wrote the manuscript. R.S. performed the experiments. R.S. and T.S. constructed the models and analyzed the data.

## Additional information

**Competing interests:** The authors declare no competing interests.

