## [Peer Review File · Nature Communications]

Reviewers' Comments:

Reviewer #1:

Remarks to the Author:

General comments

Reaction diffusion (RD) system has been proposed as one of fundamental principles to generate a variety of patterns in natural biological systems. Recent studies showed in vitro RD pattern formation with engineered bacteria, but an artificial RD circuit in mammalian cells has not yet been created. In this manuscript, the authors have reconstituted an RD circuit using Nodal and Lefty in HEK293 cells. They observed an emergent pattern and studied their system through experimental perturbation and mathematical modeling. In these studies, the authors attempted to verify that the RD circuit gave rise to a particular type of pattern and demonstrate that the reconstitution system serves as a powerful tool for analyzing how diffusion properties of Nodal and Lefty give rise to pattern formation. The authors measured the diffusion length of Nodal and Lefty and found that diffusion range of Nodal is short, confined on the basal side of cell layers due to the function of the finger 1 domain of Nodal.

In short, this manuscript illustrates a very important approach to understanding the principles of pattern formation in biological systems. However, the experimental and simulation conditions are not adequately described in some critical instances. There are also a few weak points related to the robustness of pattern formation and the conclusion described in relation to solitary pattern formation. The following revisions are required.

Major questions and recommendations;

1. The authors designed the RD circuit of Nodal and Lefty with ectopic Cryptic and FoxH1 expression in HEK293 cells. In this case, FoxH1 may induce gene expression of endogenous targets as well as the synthetic (f2)7-reporter in response to Nodal stimulation. This artificial RD circuit is interesting, but it remains unclear whether the authors built a "minimal" RD circuit. The text should be revised.

2. How robustly did patterns form from the activator-inhibitor clones?

According to the manuscript, the pattern of interest in Fig. 1 was formed by culturing a small number of luc reporter-positive cells with a large number of reporter-negative cells. What pattern would form if the experiment started under different initial conditions (e.g. 100% of reporter-positive cells or a mixture of 50% reporter-positive and 50% reporter-negative cells)? The authors should examine these questions to assess the robustness of their circuit performance in across a range of contexts. The reproducibility of pattern formation should also be tested by isolating (e.g., through cell sorting) the reporter-positive or negative population resulting from experimental pattern formation, and then observing whether the sorted cells can regenerate the pattern. The authors mentioned that the observed pattern lifetime is 50-60hr, and note this is probably due to a lack of fresh medium. Do the authors have any data supporting this discussion, or could there be other limitations of their system (e.g., luciferin substrate supply, etc) that may explain this observation?

3. Questions related to the creation of stable cell clones.

The activator and activator-inhibitor circuits generated the pattern spontaneously, so it is unclear how to keep the cells in an inactive state while the cells grew from a single cell clone. Is it necessary to maintain the cells sparsely to avoid spontaneous signal triggering? In Fig.1b and 1e, there were around 10% population of reporter-positive cells at the beginning, but were they spontaneously generated even in a clonal population during growth? In addition, when the authors built the clones, how many clones formed the pattern? Was there a variability of patterns among the clones? The information would be helpful for readers to understand how to construct and handle the synthetic signaling system.

4. Consistency of experimental results and mathematical models.

Using the mathematical models, it was shown that the competitive inhibition model yielded the

periodic Turing pattern, but competitive + direct inhibition model formed the solitary pattern. The authors suggested that their RD system may not give rise to a Turing pattern, but rather to a solitary pattern in terms of periodicity of the pattern. Here, it is critical to know how the value of k_+ (binding rate constant of Nodal/Lefty) was defined in the mathematical model. To see if the competitive + direct inhibition model is reasonable, how strongly Lefty binds to Nodal directly should be determined or known from previous work that is cited in this context.

5. Other unclear points in the figures.

In almost all figures, the patterns are shown with "Low" to "High" luminescence, but the luminescence values should be clearly stated in quantities for each figure. In some figures, the information about the size of scale bar is missing.

In Fig.2I, the apical/lateral/basal sides were imaged with confocal microscope to define the distance between them. However, the accompanying HiBiT-Nodal imaging was performed with a different microscope, LCV110. How was the basal position defined with the LCV110 when this position seems to have been defined on a different imaging system?

In supplementary figure 5, Avcr2 knock out cell line was used. The authors need to provide more detail related to how this line was established and how they validated Avcr2 was knocked out.

Additional details are required for supplementary Figure 7. Is the experimental setup the same as Fig.2a? At 3hr after the addition of Cytochalasin B, the HiBiT-Nodal signal was distributed broadly by release from the basal side, but why did most of the signal disappear at 6hr?

In supplementary figure 8b, the authors noted that the extracellular and intracellular proteins were not distinguished when determining the degradation rate of Nodal and Lefty. How were extracellular proteins including secreted proteins collected?

Reviewer #2:

Remarks to the Author:

Although reaction-diffusion (RD) system has a potential to generate patterns in developing embryos, this has never been tested experimentally with mammalian embryos. Since Nodal and Lefty act as a short-range activator and a long-range inhibitor respectively, they can compose a RD system. In this paper, the authors have designed and created a cultured cell system in which a Nodal-Lefty signaling network is reconstituted. This synthetic system was able to generate a polka-dot pattern in a culture dish, showing as the first time that the Nodal-Lefty circuit indeed forms a pattern in a living system.

One of the important requirements for an activator and inhibitor to compose a RD system is that an inhibitor must travel faster than an activator. There has been experimental data showing that Lefty indeed diffuses faster than Nodal in living embryos, but molecular features that make Nodal and Lefty short-range and long-range, respectively, remain unknown. In this paper, the authors have addressed this critical issue and have discovered a domain of Nodal that makes Nodal short-range. Thus, addition of this domain to Lefty decreased diffusivity, converting Lefty to a short-range inhibitor unable to form a pattern.

In all, this is a beautiful paper that combines synthetic biology and theoretical biology. The Nodal-Lefty system is probably responsible for various patterning events during development: scaling for example, as demonstrated by the most recent paper (Almuedo-Castillo et al, NCB 2018) showing that the Nodal-Lefty system allows scale-invariant patterning in zebrafish embryos (this paper may be mentioned in revision). The paper should be published in Nature Communication, and will receive broad interests.

Reviewer #3:

Remarks to the Author:

This paper presents the rational design of gene-expression patterns in a tissue through a combination of experiments and modeling. This is a step forward in our capabilities to rationally engineer synthetic multi-cellular systems – while mimicking natural systems. Specifically, the authors show that a pattern only emerges when two feedback-loops of activator and repressor are coupled – furthermore that the repressor needs to diffuse faster / further away – as expected from the established reaction-diffusion formalism. The authors also show that the pattern disappears when this difference in diffusivity is abolished – also as expected. The authors also explore corresponding mathematical models,

Overall the data is convincing and supports the main conclusions. I would recommend publications – I see a few issues that should be addressed.

Major:

Line 209-214: the statement that one pattern is highly periodic and the other is less so. I don't feel that is the right statement. When I look at the corresponding simulation results I see in both cases patterns – each with some typical intrinsic length scale - some consisting of dots only, some of stripes and dots, and some more strips (with additional branching). But I don't see any quantitative measure of difference in periodicity. The authors should either reword or measure periodicity.

For the parameters used for the model (lines 414-419) – it should be stated where these number come from – either reference to own measurement of an parameter, or literature - or after fitting solutions to experimental results. For example, the Hill coefficients of 2.63 and 1.09 are strangely specific without any motivation. Also – some information should be given how robust these results qualitatively are given changes in parameters

Fig. 1 b,c vs. e,f,g: Some additional quantitative tests are needed: One could argue from the images that the pattern in Fig.1b30h does not look too different from Fig.1e42h (as an example). And while Fig.1b42h is green-red while Fig.1e4242 is more blue-green – in both cases one could argue that there are patterns there with similar feature sizes. Hence the authors should provide some quantitative analysis, for example to measure the “contrast,” i.e., difference between max and min intensity (potentially also normalizing by the average intensity) – or subtract average intensity / normalize by average intensity.

Regarding data Fig.e,f,g: For inhibition one would expect that the regions that ultimately end up low were initially increasing before decreasing again. It would be interesting to reanalyze the data to check that in a time course – and also compare to the regions that end up high. Potentially one could see even some temporal oscillations in some regions. This analyses would likely also require some suitable binning.

Minor:

Do the authors also have a movie example without these circuits, specifically what is shown in Supp Figs. 1 and 2. Having that would be nice for comparison. (Especially the condition in Supp Fig.1 would be helpful to compare / understand what the baseline is) (if no movie is available – repeating experiments might not be needed)

Fig.1: Scale bar description missing

The last sentence (“the first step”) feels like an overclaim – as there other examples in the literature (see recent paper from Wendell Lim lab – but also others) – I suggest softening that.

It would be nice to read in the discussion a bit more about how this informs our understand of nodal and lefty in natural systems.

Fig2e: curve fitting – misses information on SEM; N

Improve the panel alignment / overall space usage in Fig.2 (similar for Fig.4)

In reply to Reviewer #1

“General comments

Reaction diffusion (RD) system has been proposed as one of fundamental principles to generate a variety of patterns in natural biological systems. Recent studies showed in vitro RD pattern formation with engineered bacteria, but an artificial RD circuit in mammalian cells has not yet been created. In this manuscript, the authors have reconstituted an RD circuit using Nodal and Lefty in HEK293 cells. They observed an emergent pattern and studied their system through experimental perturbation and mathematical modeling. In these studies, the authors attempted to verify that the RD circuit gave rise to a particular type of pattern and demonstrate that the reconstitution system serves as a powerful tool for analyzing how diffusion properties of Nodal and Lefty give rise to pattern formation. The authors measured the diffusion length of Nodal and Lefty and found that diffusion range of Nodal is short, confined on the basal side of cell layers due to the function of the finger 1 domain of Nodal.

In short, this manuscript illustrates a very important approach to understanding the principles of pattern formation in biological systems. However, the experimental and simulation conditions are not adequately described in some critical instances. There are also a few weak points related to the robustness of pattern formation and the conclusion described in relation to solitary pattern formation. The following revisions are required.

Major questions and recommendations;

1. The authors designed the RD circuit of Nodal and Lefty with ectopic Cryptic and FoxH1 expression in HEK293 cells. In this case, FoxH1 may induce gene expression of endogenous targets as well as the synthetic (f2)7-reporter in response to Nodal stimulation. This artificial RD circuit is interesting, but it remains unclear whether the authors built a “minimal” RD circuit. The text should be revised.”

As the reviewer pointed out, our artificial gene circuit might affect endogenous target genes of FoxH1. Thus, the word ‘minimal’ was removed from the entire text.

“2. How robustly did patterns form from the activator-inhibitor clones?

According to the manuscript, the pattern of interest in Fig. 1 was formed by culturing a small number of luc reporter-positive cells with a large number of reporter-negative cells. What pattern would form if the experiment started under different initial conditions (e.g. 100% of reporter-positive cells or a mixture of 50% reporter-positive and 50% reporter-negative cells)? The authors should examine these questions to assess the robustness of their circuit performance in across a range of contexts.”

In the previous pattern formation experiments, we did not mix reporter-positive cells and negative cells: the (clonal) cells were simply re-split at a near confluent density, and then about 10% of the cells

happened to be positive initially. Now, according to the reviewer's suggestion, we performed co-culture experiments to manipulate the initial condition (new Supplementary fig. 9): the positive cells and negative cells were prepared by pretreatment of the cells with recombinant Nodal and the inhibitor SB431542, respectively, and they were mixed at different ratios. Interestingly, the resulting pattern significantly varied according to the initial mix ratio, suggesting that our pattern may be a solitary pattern, which depends on the initial condition, rather than a Turing pattern, which is independent of the initial condition. We thank the reviewer for the insightful suggestion.

“The reproducibility of pattern formation should also be tested by isolating (e.g., through cell sorting) the reporter-positive or negative population resulting from experimental pattern formation, and then observing whether the sorted cells can regenerate the pattern.”

Because we used a luminescent reporter, instead of a fluorescent one, cell sorting with FACS is impossible. Thus, we re-cloned the pattern forming cell line from a single cell, confirming that the re-cloned cells formed an essentially the same pattern (new Fig. 1h).

“The authors mentioned that the observed pattern lifetime is 50-60hr; and note this is probably due to a lack of fresh medium. Do the authors have any data supporting this discussion, or could there be other limitations of their system (e.g., luciferin substrate supply, etc) that may explain this observation?”

As shown above, changing the culture medium at 66 hour increased the luciferase reporter signal (only transiently, though), and so we think that at least fresh medium is important. However, this signal recovery can be because of new luciferin supply, as the reviewer indicated. Thus, we have revised the text: ‘The propagated reporter signal lasted until 50-60 hours and started to gradually decline possibly due to the lack of fresh medium and/or luciferin supply’.

“3. Questions related to the creation of stable cell clones.

The activator and activator-inhibitor circuits generated the pattern spontaneously, so it is unclear how to keep the cells in an inactive state while the cells grew from a single cell clone. Is it necessary to

maintain the cells sparsely to avoid spontaneous signal triggering? In Fig.1b and 1e, there were around 10% population of reporter-positive cells at the beginning, but were they spontaneously generated even in a clonal population during growth?"

Passaging cells resets and decreases the reporter signal possibly because the Nodal molecules confined between the cells and the culture dish are lost through re-splitting. Thus, it is not necessary to maintain the cells sparsely: the cells were regularly maintained at a subconfluent state and re-split at a fixed density immediately before pattern formation experiments, resulting in ~10% of positive cells. In addition, re-cloning of the pattern forming cell line from a single cell reproduced an essentially the same pattern (new Fig. 1h), suggesting that the initial 10% positive cells may be spontaneously generated.

"In addition, when the authors built the clones, how many clones formed the pattern? Was there a variability of patterns among the clones? The information would be helpful for readers to understand how to construct and handle the synthetic signaling system."

This is an important point. We added the information of cell clones to new Methods section: 'As for the activator circuit, 5 out of 30 clonal lines showed successful signal propagation, and 1 line was used as a base for the activator-inhibitor circuit. As for the activator-inhibitor circuit, 2 out of 80 clonal lines showed successful cell patterning, and 1 line was rigorously analyzed'. The 2 activator-inhibitor lines showed slightly different patterns due to the differences in the copy number and the genome integration site of genetic constructs. The fact that different copy numbers of genetic constructs give rise to a variety in the patterns is also exemplified in Fig. 3n, where the copy numbers were explicitly increased.

"4. Consistency of experimental results and mathematical models.

Using the mathematical models, it was shown that the competitive inhibition model yielded the periodic Turing pattern, but competitive + direct inhibition model formed the solitary pattern. The authors suggested that their RD system may not give rise to a Turing pattern, but rather to a solitary pattern in terms of periodicity of the pattern. Here, it is critical to know how the value of k_+ (binding rate constant of Nodal/Lefty) was defined in the mathematical model. To see if the competitive + direct inhibition model is reasonable, how strongly Lefty binds to Nodal directly should be determined or known from previous work that is cited in this context."

Because it is difficult to directly measure the binding constant of Nodal and Lefty (k_+), we varied the value of k_+ over 4 orders of magnitude in simulations (new Supplementary fig. 8). Solitary patterns were formed as long as k_+ was relatively small ($< 0.3 \text{ min}^{-1}\text{nM}^{-1}$). In addition, the fact that our pattern

significantly varies according to the initial condition (new Supplementary fig. 9) also suggests that our pattern may be a solitary pattern rather than a Turing pattern.

“5. Other unclear points in the figures.

In almost all figures, the patterns are shown with “Low” to “High” luminescence, but the luminescence values should be clearly stated in quantities for each figure. In some figures, the information about the size of scale bar is missing.”

The values were added to all the color bars. The scale bar information was added to the legend of Fig. 1.

“In Fig.2l, the apical/lateral/basal sides were imaged with confocal microscope to define the distance between them. However, the accompanying HiBiT-Nodal imaging was performed with a different microscope, LCV110. How was the basal position defined with the LCV110 when this position seems to have been defined on a different imaging system?”

Both images of mCherry-CAAX and HiBiT-Nodal (Fig. 2l,m) were taken by the same microscope, LCV110, and the basal position was defined using the mCherry-CAAX image. The confocal images (Supplementary fig. 6) were used just as additional information to show higher resolution images of cell morphologies at basal/apical sides. To avoid the confusion, we have cited Fig. 2l,m when explaining the definition of basal/apical sides: ‘The basal side was judged with the dense structure of cell membrane, and the lateral and apical sides were defined as the points 7.5 μm and 15 μm above the basal side, respectively (Fig. 2l,m; for higher resolution images, see Supplementary fig. 6)’.

“In supplementary figure 5, Avcr2 knock out cell line was used. The authors need to provide more detail related to how this line was established and how they validated Avcr2 was knocked out.”

According the reviewer’s suggestion, we added the PCR bands and additional information of the Avcr2b deletion mutant (new Supplementary fig. 5b). Also, the PCR primer information was added to new Supplementary table 1.

“Additional details are required for supplementary Figure 7. Is the experimental setup the same as Fig.2a? At 3hr after the addition of Cytochalasin B, the HiBiT-Nodal signal was distributed broadly by release from the basal side, but why did most of the signal disappear at 6hr?”

As the reviewer pointed out, the HiBiT-Nodal signal disappeared at 6 hours after Cytochalasin B treatment for some unknown reasons. Because Cytochalasin B inhibits actin polymerization that affects many cellular processes, it is difficult to tell exactly what caused the change of Nodal distribution. Thus, we have realized this data is too premature and decided to get rid of it from the manuscript.

“In supplementary figure 8b, the authors noted that the extracellular and intracellular proteins were not distinguished when determining the degradation rate of Nodal and Lefty. How were extracellular proteins including secreted proteins collected?”

To measure the degradation rate of intracellular/extracellular proteins, we just scraped and lysed the adherent HEK293 cells, ignoring the extracellular proteins floating in the medium. This is partly justified by the fact that the extracellular Nodal and Lefty that contribute to pattern formation are not floating but trapped around the cells: the medium change does not disturb the extracellular distribution or pattern (see the pictures shown above in response to the comment ‘*The authors mentioned that the observed pattern lifetime is 50-60hr, and note this is probably due to a lack of fresh medium...*’). We have explained the limitation of the measurement in the legend of new Supplementary fig. 7b: ‘Note that the extracellular and intracellular proteins were not distinguished and that extracellular proteins floating in the medium were ignored in this measurement’.

In reply to Reviewer #2

“Although reaction-diffusion (RD) system has a potential to generate patterns in developing embryos, this has never been tested experimentally with mammalian embryos. Since Nodal and Lefty act as a short-range activator and a long-range inhibitor respectively, they can compose a RD system. In this paper, the authors have designed and created a cultured cell system in which a Nodal-Lefty signaling network is reconstituted. This synthetic system was able to generate a polka-dot pattern in a culture dish, showing as the first time that the Nodal-Lefty circuit indeed forms a pattern in a living system.

One of the important requirements for an activator and inhibitor to compose a RD system is that an inhibitor must travel faster than an activator. There has been experimental data showing that Lefty indeed diffuses faster than Nodal in living embryos, but molecular features that make Nodal and Lefty short-range and long-range, respectively, remain unknown. In this paper, the authors have addressed this critical issue and have discovered a domain of Nodal that makes Nodal short-range. Thus, addition of this domain to Lefty decreased diffusivity, converting Lefty to a short-range inhibitor unable to form a pattern.

In all, this is a beautiful paper that combines synthetic biology and theoretical biology. The Nodal-Lefty system is probably responsible for various patterning events during development: scaling for example, as demonstrated by the most recent paper (Almuedo-Castillo et al, NCB 2018) showing that the Nodal-Lefty system allows scale-invariant patterning in zebrafish embryos (this paper may be mentioned in revision). The paper should be published in Nature Communication, and will receive broad interests.”

According to the reviewer’s suggestion, we briefly mentioned the role of the differential diffusivity of Nodal and Lefty in the scaling of zebrafish embryos and cited the paper from Almuedo-Castillo et al (new Discussion section): ‘Very recently, the differential diffusivity of Nodal and Lefty has also been reported to underlie scaling of the proportions of germ layers in zebrafish’.

In reply to Reviewer #3

“This paper presents the rational design of gene-expression patterns in a tissue through a combination of experiments and modeling. This is a step forward in our capabilities to rationally engineer synthetic multi-cellular systems – while mimicking natural systems. Specifically, the authors show that a pattern only emerges when two feedback-loops of activator and repressor are coupled – furthermore that the repressor needs to diffuse faster / further away – as expected from the established reaction-diffusion formalism. The authors also show that the pattern disappears when this difference in diffusivity is abolished – also as expected. The authors also explore corresponding mathematical models,

Overall the data is convincing and supports the main conclusions. I would recommend publications – I see a few issues that should be addressed.

Major:

Line 209-214: the statement that one pattern is highly periodic and the other is less so. I don’t feel that is the right statement. When I look at the corresponding simulation results I see in both cases patterns – each with some typical intrinsic length scale - some consisting of dots only, some of stripes and dots, and some more strips (with additional branching). But I don’t see any quantitative measure of difference in periodicity. The authors should either reword or measure periodicity.”

We figured that a clearer criteria that distinguish a solitary pattern from a Turing pattern should be necessary. Thus, we altered the initial condition in the pattern formation experiment by co-culturing the reporter-positive cells and negative cells at different mix ratios (new Supplementary fig. 9). Because the

resulting pattern varied according to the initial condition, we concluded that our cellular pattern resembles a solitary pattern, which depends on the initial condition, rather than a Turing pattern, which is independent of the initial condition. We also weakened the expression ‘highly periodic’: ‘The domains resulting from the competitive inhibition + direct inhibition model showed less regular size and shape (Fig. 3h,i) compared with those of the Turing pattern’. We thank the reviewer for the insightful suggestion.

“For the parameters used for the model (lines 414-419) – it should be stated where these number come from – either reference to own measurement of an parameter, or literature - or after fitting solutions to experimental results. For example, the Hill coefficients of 2.63 and 1.09 are strangely specific without any motivation. Also – some information should be given how robust these results qualitatively are given changes in parameters”

This is an important point. We have summarized the source of parameter values in new Methods section. The Hill coefficients are specific because they were estimated by parameter fitting. Because the maximum production rate (α) and the association rate of Nodal and Lefty (k_+) were arbitrarily chosen, we varied these parameters to examine the pattern forming range (new Supplementary fig. 8).

“Fig. 1 b,c vs. e,f,g: Some additional quantitative tests are needed: One could argue from the images that the pattern in Fig.1b30h does not look too different from Fig.1e42h (as an example). And while Fig.1b42h is green-red while Fig.1e4242 is more blue-green – in both cases one could argue that there are patterns there with similar feature sizes. Hence the authors should provide some quantitative analysis, for example to measure the “contrast,” i.e., difference between max and min intensity (potentially also normalizing by the average intensity) – or subtract average intensity / normalize by average intensity.”

According to the reviewer’s suggestion, we compared the spatial correlations of ‘activator 30 h’, ‘activator 42 h’, and ‘activator-inhibitor 42 h’ (new Supplementary fig. 4). Among the three, only activator-inhibitor 42 h showed a small second peak in the spatial correlation, indicating that pattern formation is possible only with the activator-inhibitor circuit.

”Regarding data Fig.e,f,g: For inhibition one would expect that the regions that ultimately end up low were initially increasing before decreasing again. It would be interesting to reanalyze the data to check that in a time course – and also compare to the regions that end up high. Potentially one could see even

some temporal oscillations in some regions. This analyses would likely also require some suitable binning.”

As shown in the kymographs above, some regions indeed displayed a transient increase in the reporter signal before ending up being low. Although the fluctuation in the reporter signal is quite interesting, such regions are rather rare, and we currently lack a thorough analysis or explanation of the fluctuation. Thus, we decided not to discuss it in the manuscript.

“Minor:

Do the authors also have a movie example without these circuits, specifically what is shown in Supp Figs. 1 and 2. Having that would be nice for comparison. (Especially the condition in Supp Fig.1 would be helpful to compare / understand what the baseline is) (if no movie is available – repeating experiments might not be needed)”

According to the reviewer’s suggestion, we added a time course of cells without any circuits (new Supplementary fig. 1c): the reporter cells treated with recombinant Nodal showed a homogeneous increase in the reporter signal but did not give rise to any pattern as expected.

“Fig.1: Scale bar description missing”

The scale bar information was added to the legend of Fig. 1.

“The last sentence (“the first step”) feels like an overclaim – as there other examples in the literature (see recent paper from Wendell Lim lab – but also others) – I suggest softening that.”

Actually, our original intention was to emphasize that this work is ‘only a first step’ towards a more complex synthetic tissue. Thus, we have revised the sentence: ‘This work will also serve as a base for engineering a more complex synthetic tissue’.

“It would be nice to read in the discussion a bit more about how this informs our understand of nodal and lefty in natural systems.”

A recent report on the role of the differential diffusivity of Nodal and Lefty in the scaling of zebrafish embryos (Almuedo-Castillo et al, Nat Cell Biol, 2018) was briefly discussed (new Discussion section): ‘Very recently, the differential diffusivity of Nodal and Lefty has also been reported to underlie scaling of the proportions of germ layers in zebrafish’.

Fig2e: curve fitting – misses information on SEM; N

The information of SEM and N was added to the legend of Fig. 2e,h,k.

Improve the panel alignment / overall space usage in Fig.2 (similar for Fig.4)”

We have rearranged the panels of Figs. 2 and 4 to save the space.

Other changes:

- To meet the word limit, the abstract and topical headings were shortened.
- To meet the publisher’s guideline, the display of FACS data was changed from a histogram to a dot plot (new Fig. 3m), and gene names were italicized.

Our manuscript has been greatly improved by the reviewers’ constructive comments. Thank you all very much.

Reviewers' Comments:

Reviewer #1:

Remarks to the Author:

The authors have addressed the questions in my review comments and significantly improved the manuscript. For example, by testing various initial conditions of ratios of positive cells and negative cells, they show that the pattern is dependent on the initial conditions, which suggest the pattern may be a solitary pattern rather than a Turing pattern. I have following two questions based on the response letter, but, otherwise, this work is suitable for publication in Nature Communications.

The authors added the information about cell clones. They used one clone of the activator circuit as a base to build the activator-inhibitor circuit and acquired 2 successful clones of the activator-inhibitor circuit out of 80 clones. I'm wondering if the authors could also provide information about how many clones were tested for similar range activator-inhibitor circuit that induces F1-Lefty2 in Fig.4g.

The experimental results suggest that the pattern is a solitary pattern, but the solitary pattern is dependent on the direct binding of Nodal and Lefty. If it is difficult to measure the association rate of Nodal and Lefty (the value of k_+), the authors should mention that the association rate of Nodal and Lefty remains to be determined somewhere in the paragraph of discussion about solitary pattern vs Turing pattern (line 298~308).

Reviewer #3:

Remarks to the Author:

Regarding new Supplementary Fig.4: The authors need to provide statistical tests that the relevant features in the curves b,d,f are significantly different.

In reply to Reviewer #1

“The authors have addressed the questions in my review comments and significantly improved the manuscript. For example, by testing various initial conditions of ratios of positive cells and negative cells, they show that the pattern is dependent on the initial conditions, which suggest the pattern may be a solitary pattern rather than a Turing pattern. I have following two questions based on the response letter, but, otherwise, this work is suitable for publication in Nature Communications.

The authors added the information about cell clones. They used one clone of the activator circuit as a base to build the activator-inhibitor circuit and acquired 2 successful clones of the activator-inhibitor circuit out of 80 clones. I’m wondering if the authors could also provide information about how many clones were tested for similar range activator-inhibitor circuit that induces FI-Lefty2 in Fig.4g.”

According to the suggestion, we have explained the number and phenotype of clones of the similar range activator-inhibitor circuit in the Method section: ‘As for the similar range activator-inhibitor circuit, 8 out of 12 clonal lines analyzed showed bright homogeneous images (represented by clone 1 in Fig. 4g), and 4 lines showed dark homogeneous images (represented by clone 2)’.

Additionally, we have corrected the numbers of clones of the activator circuit or activator-inhibitor circuits. We previously wrote ‘As for the activator circuit, 5 out of 30 clonal lines showed successful signal propagation’ and ‘As for the activator-inhibitor circuit, 2 out of 80 clonal lines showed successful cell patterning’. However, these numbers (30 and 80) include the clones that might not contain the necessary DNA construct. Since these descriptions are rather confusing, we have decided to use the numbers of clones that were tested-positive for the genome integration of the construct and analyzed for pattern forming ability with a microscope. Here are the new descriptions: ‘As for the activator circuit, 5 out of 6 clonal lines analyzed showed successful signal propagation’; ‘As for the activator-inhibitor circuit, 2 out of 9 clonal lines analyzed showed successful cell patterning’.

“The experimental results suggest that the pattern is a solitary pattern, but the solitary pattern is dependent on the direct binding of Nodal and Lefty. If it is difficult to measure the association rate of Nodal and Lefty (the value of k_+), the authors should mention that the association rate of Nodal and Lefty remains to be determined somewhere in the paragraph of discussion about solitary pattern vs Turing pattern (line 298~308).”

According to the suggestion, we have added an explanation to the Discussion section: ‘Although the fact that our cell pattern significantly varied according to the initial condition favors a solitary pattern over a Turing pattern, further experiments are needed to distinguish the two possibilities, including the

direct measurement of the association rate of Nodal and Lefty in the competitive inhibition + direct inhibition model’.

In reply to Reviewer #3

“Regarding new Supplementary Fig.4: The authors need to provide statistical tests that the relevant features in the curves b,d,f are significantly different.”

According to the suggestion, we have performed a statistical test to compare the spatial correlations of Activator-inhibitor 48 h and Activator 30 h. We explained the result in the manuscript: ‘Although the image of the activator circuit in the middle of the signal propagation process showed positive domains and negative domains (Fig. 1b, 30 h), its correlation did not show a clear second peak (Supplementary fig. 4f; Activator-inhibitor 48 h vs. Activator 30 h, $p = 0.029$, Wilcoxon rank sum test).’. We also explained the statistical method in new Supplementary Fig. 4: ‘To perform a statistical test, the integral of radial correlation between 150 - 250 μm was subtracted from the integral of radial correlation between 300 - 400 μm . The calculated value for Activator 48 h (b), Activator-inhibitor 48 h (d) and Activator 30 h (f) was -1.88 ± 0.55 , 1.75 ± 0.74 and -1.16 ± 0.37 , respectively ($n = 4$, mean \pm s.e.m). The two-sided Wilcoxon rank sum test was performed to compare Activator-inhibitor 48 h and Activator 30 h ($p = 0.029$).’.